# Adult-Onset Familial Hemophagocytic Lymphohistiocytosis Presenting with Annular Erythema following COVID-19 Vaccination

**DOI:** 10.3390/vaccines10091436

**Published:** 2022-08-31

**Authors:** Yifan He, Yun Hui, Haibo Liu, Yifan Wu, Hong Sang, Fang Liu

**Affiliations:** 1Department of Dermatology, Jinling Hospital, Nanjing Medical University, Nanjing 210002, China; 2Department of Dermatology, Jinling Hospital, Nanjing University, Nanjing 210002, China

**Keywords:** hemophagocytic lymphohistiocytosis, cutaneous manifestations, COVID-19 vaccination, gene mutation

## Abstract

Familial hemophagocytic lymphohistiocytosis (HLH) is a rare genetic and life-threatening immunodeficiency disease. Here, we present a 38-year-old male who initially developed multiple annular to irregular erythema accompanied by a fever after COVID-19 vaccination. He was diagnosed with HLH with evidence of leukocytopenia in a full blood test, elevations of ferritin and sCD25, decreased NK cell function, and hemophagocytosis of a bone marrow biopsy specimen. A genetic examination revealed two probable disease-causing heterozygous mutations on UNC13D associated with type 3 familial HLH. A review of the case reports relevant to HLH following COVID-19 vaccination and the cutaneous manifestations of HLH with genetic defects suggests the necessity that individuals with preexisting immune dysregulation or diseases not classified should be cautious about COVID-19 vaccination and reminds clinicians that various recalcitrant skin lesions may be a sign of HLH.

## 1. Introduction

Hemophagocytic lymphohistiocytosis (HLH) is a rare and life-threatening immune-mediated syndrome characterized by uncontrolled and persistent activation of cytotoxic T lymphocytes and natural killer (NK) cells, leading to overwhelming systemic inflammation [1]. According to the etiologies, HLH is often categorized as primary and secondary/reactive HLH. Primary HLH is caused by genetic defects affecting the cytotoxic function of T lymphocytes and NK cells and normally presents in young children [1,2]. Primary HLH, including familial HLH and related immune disorders associated with an HLH phenotype, have been defined as autosomal recessive diseases. Most pediatric patients have identifiable genetic defects inherited in a Mendelian fashion as homozygous or compound heterozygous lesions [1,3]. Secondary HLH, generally affecting adolescents and adults, is always associated with diseases causing immune dysregulation in the absence of an identifiable underlying genetic mutation, such as infection, malignancy (particularly lymphoma), autoimmune conditions, or other circumstances [2,4].

HLH is primarily triggered by herpes and EB viruses but may be associated with other viral, bacterial, fungal, and parasitic infections or some non-infectious triggers such as drugs, vaccinations, surgery, and severe burns [5]. HLH induced by vaccination is uncommon. With the COVID-19 pandemic, the emerging SARS-CoV-2 vaccines have been widely applied in epidemic prevention and control. Although the benefits of the vaccines are undoubted, the adverse effects may be neglected and deserve more studies to elaborate on this topic.

Herein, we report a 38-year-old male with an initial presentation of widespread annular erythema on the trunk following SARS-CoV-2 vaccination, who was subsequently diagnosed with familial HLH. This is the first case of HLH triggered by COVID-19 vaccination immunostimulation on a genetic defect of a UNC13D background. 

## 2. Case Report 

A 38-year-old man was presented to our department with multiple pruritic annular to irregular erythema on the trunk for 2 months. He had received the COVID-19 vaccination approximately 4 weeks before the appearance of skin rashes. The patient suffered from severe interstitial pneumonia of unknown causes 3 years ago and was in stable condition in recent years. He was suspected of urticarial vasculitis (UV), erythema annulare centrifugum (EAC), or eosinophilic annular erythema (EAE). However, the histopathology indicated perivascular and periappendage infiltration comprising histocytes and atypical lymphocytes, with an obvious involvement of nerves (Figure 1). Meanwhile, complete blood count, immune serum globulin, complement and complete autoimmune antibody tests, including anti-nuclear antibody (ANA) spectrum, antibodies to double-stranded DNA (anti-dsDNA) antibody, anti-neutrophil cytoplasmic antibodies (ANCA), anti-cardiolipin antibody (ACA), rheumatoid factor (RF), etc., were all at normal levels. Thus, the diagnoses of UV, EAC, and EAE were excluded. Given the involvement of nerves in the pathologic findings, the possibility of leprosy was taken into account. However, immunofluorescence, acid-fast staining, enzyme-linked immunospot assay (ELISPOT), and next-generation sequencing (NGS) all showed no evidence of mycobacterium leprae infection. He was then treated with antihistamines, glucocorticosteroid, and thalidomide, with mild alleviation of the lesions. 

However, the lesions spread rapidly shortly after the patient received the booster shot of the COVID-19 vaccine, presenting with generalized pruritic annular to irregular erythema, accompanied by edema on his left face (Figure 2). Almost one month later, new accompanying symptoms, including a high fever and fatigue, appeared. A whole-blood test indicated leukocytopenia (WBC 2.09 × 10^9^/L). Marrow aspiration was performed, and the result revealed the phagocytosis of platelets, erythrocytes, granulocytes, and other granular fragments by a number of hemophagocytes in the bone marrow smears. Meanwhile, he had dramatically elevated levels of ferritin and sCD25 and decreased NK cell function. The results of the laboratory tests are summarized in Table 1. Based on the HLH-2004 diagnostic criteria, a diagnosis of hemophagocytic lymphohistiocytosis was confirmed. A coherent timeline indicating the date of this case is presented in Figure 3.

It is known that infection, malignancy (especially lymphoma), autoimmune disease, and gene deficiency are common etiologies of HLH. Therefore, a series of examinations were carried out to explore the underlying causes. Hepatitis B and C virus (HBV and HCV) screening, HIV screening, and a syphilis serum test were all negative, and NGS detected no pathogenic microorganisms in skin and blood specimens. Positron emission tomography/computed tomography (PET/CT) revealed widespread increased metabolic activity (Figure 4). A repeated skin biopsy was performed to exclude the diagnosis of lymphoma. The histopathology revealed lymphohistiocytic infiltration in the dermis as well as reactive hyperplasia of lymphoid tissue dominated by cytotoxic T-cells. Immunohistochemistry showed that these cells were positive for CD3, CD2, CD5, CD7, CD4, CD43, TIA, GrB, and perforin, weak-positive for CD8, P53, C-myc, Ki-67 (20–30%), and negative for CD21, CD23, CD20, CD56, MUM1, cyclinD1, CD10, Bc16, PAX5, CD30, PD1, CD38, kappa, lambda, and EBER. Lymphoma was also excluded based on the existing evidence, while genetic examination revealed two probable disease-causing heterozygous mutations on UNC13D, c.817C > T (p.R273*) and c.2588G > A (p.G863D), respectively. Therefore, the gene mutation was considered the cause of HLH. Furthermore, his mother was found to be a carrier of the UNC13D (p.R273*) mutation, and his father was found to be a carrier of the UNC13D (p.G863D) mutation. Finally, this patient was diagnosed as familial HLH type 3, with a mutation of UNC13D triggered by COVID-19 vaccination. The patient was given the therapy of an allogeneic bone marrow transplant.

## 3. Literature Review

### 3.1. HLH following COVID-19 Vaccination

English language cases of HLH following COVID-19 vaccination were reviewed and analyzed using the PubMed database from the date of database inception until 20 June 2022; “(COVID-19 OR SARS-CoV-2) AND (vaccine OR vaccination) AND (hemophagocytic lymphohistiocytosis OR hemophagocytic syndrome)” was used as a search filter.

A total of 12 pieces of literature with 20 HLH cases are included and summarized in Table 2, of which 12 cases were males (60.0%) with a mean age of onset of 48.10 years (14–85 years); 6/20 (30.0%) cases were accompanied by skin manifestations, including maculopapules, pruritus, jaundice, and facial swelling. Our case is the only one that presented with isolated skin lesions initially. The mean interval between onset symptoms and vaccination was 9.31 days (1–28 days); 10/20 (50.0%) patients had underlying medical conditions such as interstitial lung disease, HIV, adult-onset Still’s disease, diabetes mellitus, malignancy, heart failure; 4/20 (20.0%) patients were positive for EBV infection. Only one case explored the predisposing gene mutation without a positive result, and our patient is the first one who has tested positive for genetic mutations. The main therapy was corticosteroid in 17 out of 19 patients (89.47%). The common combination treatments included intravenous immunoglobulins (IVIG) (6/19, 31.58%), etoposide (4/19, 21.05%), anakinra (4/19, 21.05%), rituximab (1/19, 5.26%), infliximab (1/19, 5.26%) and emapalumab-lzsg (1/19, 5.26%). The conditions of 15/19 (78.95%) patients improved after multiple treatments, but two of them relapsed and got worse after discharge; 3/19 (15.79%) patients died of HLH or complications. The patients with other systemic conditions had worse prognoses. Only one improved spontaneously with no therapy due to clinical stability and improvement of HLH parameters.

### 3.2. Cutaneous Manifestations of HLH with Genetic Defects

English language cases of cutaneous manifestations of HLH with genetic defects were identified and analyzed using the PubMed database from the date of database inception until 20 June2022; “(skin OR cutaneous) AND (hemophagocytic lymphohistiocytosis OR hemophagocytic syndrome)” was used as a search filter. 

A total of 19 case reports are shown in Table 3, of which 11 cases were males (57.89%) with ages of onset ranging from 6 days to 30 years old; 3/19 (15.80%) patients presented with initially isolated skin lesions. The subtypes of Griscelli syndrome type 2 and Chediak–Higashi syndrome were characterized by cutaneous hypopigmentation, while cutaneous manifestations were nonspecific to other subtypes. The nonspecific skin manifestations included edema (3/19, 15.80%), widespread maculopapular rash (2/19, 10.53%), papulopustules (2/19, 10.53%), granuloma (2/19, 10.53%), erythema nodosum (1/19, 5.26%), panniculitis (2/19, 10.53%), violaceous nodules (1/19, 5.26%), ecchymosis (1/19, 5.26%), and hemorrhagic skin eruptions (1/19, 5.26%). The histologic appearances were nonspecific and rarely demonstrated cutaneous hemophagocytosis. The main treatments were induction therapy, including corticosteroid (13/15, 86.67%), etoposide (7/15, 46.67%), and cyclosporine (8/15, 53.33%); 7/15 (46.67%) patients underwent or had planned to undergo hematopoietic stem cell transplantation (HSCT). Additionally, other treatments, including IVIG (1/15, 6.67%), mycophenolate mofetil (1/15, 6.67%), and 13-cis retinoic acid (1/15, 6.67%), were applied to patients. The conditions of 6 (31.58%) cases improved after multiple treatments, 7 (36.84%) cases died of HLH or complications, and 6 cases did not provide the outcomes.

## 4. Discussion

Hemophagocytic lymphohistiocytosis (HLH) is a rare, life-threatening hyperinflammatory syndrome characterized by an uncontrollable immune response, resulting in increased secretion of inflammatory cytokines and macrophage activation [1]. Primary HLH occurs in the presence of an underlying predisposing genetic defect damaging the immune function. The genetic defects causing primary HLH are listed in Table 4. Two probable disease-causing heterozygous mutations on UNC13D were identified in the present case, which confirmed the diagnosis of F-HLH type 3 (FH3). It is known that UNC13D encodes the protein Munc13-4, which participates in the regulation of cytolytic granule maturation and exocytosis. The absence of Munc13-4 often causes the impairment of cytotoxic activity in T lymphocytes, leading to sequential reactions of inability to clear antigenic stimulus, the exuberant production of cytokines, and infiltration with hyperactivated macrophages and histiocytes in organ systems [1,37]. 

Notably, the patient provided an important fact that the lesions were stirred up after the first dose of the COVID-19 vaccine; he deteriorated significantly after the booster shot. Therefore, vaccination was considered the trigger of the present HLH patient’s underlying genetic defect by excluding other precipitating factors relevant to HLH. The mechanisms of COVID-19-vaccine-related HLH have not been illuminated. We hypothesize that increases in proinflammatory cytokines resulting from the vaccination provide an initial stimulus for the activation of cytotoxic T lymphocytes and NK cells; the impaired cytotoxic function in T lymphocytes, due to the genetic defect, leads to an inability to eliminate the antigenic stimulus, which then eventually results in a persistent and amplifying immune response [1,38]. 

Only 1 of the 20 HLH patients (following COVID-19 vaccination) from the reviewed literature explored the predisposing gene mutation but without positive results. The present patient was the first reported case with underlying genetic defects. Some scholars have proposed that the screening sequencing of HLH-associated genes is unnecessary in adult HLH cases as genetic predisposition plays a primary role in pediatric settings [39]. However, 42.9% (48/112) of patients harbored pathogenic gene mutations or rare variants in adults with HLH in Miao’s research (Miao et al. [40]). With the widespread availability of genetic testing, HLH has been described to occur throughout life, from in utero to the seventh decade [41]. Miao et al. found nonsense and frame-shift mutations were prevalent in pediatric cases, while almost all the variants were missense mutations in adult cases. Hence, he speculated that most gene mutations in adult HLH may partly impair rather than eliminate the function of the involved proteins [40]. HLH is susceptible in individuals with gene mutations after the stimulation of various triggers or predisposing disorders. Therefore, the COVID-19 vaccine may have played a stimulating role in the present HLH patient with genetic defects of UNC13D. The present case had a history of severe interstitial pneumonia of unknown causes, and the skin lesions had not been classified. Therefore, individuals with preexisting immune dysregulation or diseases not classified should be cautious about vaccination. Gene sequencing may be considered to exclude the underlying predisposing genetic defect before vaccination in some circumstances.

The initial symptoms of HLH are variable and nonspecific, leading to difficulty in early diagnosis. The clinical presentations include a continuous high fever (>38.5 °C) of unknown origin, hepatosplenomegaly, cytopenias, skin rashes, acute liver failure, bleeding, and inflammatory central nervous system disease [42]. Cutaneous manifestations are variable and nonspecific, with an estimated prevalence of 24–40% in primary HLH and 6–65% in secondary HLH, which can be easily neglected by clinicians [43]. The current case was suspected of HLH until he presented manifestations of fever and leukocytopenia and was misdiagnosed with several diseases such as urticarial vasculitis, erythema annulare centrifugum, and leprosy for the nonspecific symptoms at the earlier disease stage. Therefore, the characteristic of cutaneous manifestations of HLH represents a diagnostic challenge for clinicians, which may provide a clue for earlier diagnosis. Fardet et al. [44] summarized that the cutaneous manifestations of reactive HLH mainly fell into three types: (i) specific to the underlying malignancy (e.g., cutaneous lymphoma, systemic disease), (ii) reflecting biological consequences of HLH (e.g., conjunctival jaundice, thrombogenic purpura), and (iii) a generalized, transient, nonpruriginous maculopapular rash. In addition, generalized erythroderma, edema, nonhealing ulcers, panniculitis, and Kawasaki-like abnormalities have also been reported to occur in HLH [43]. 

To explore the characteristics of cutaneous manifestations in HLH with genetic defects, we have summarized previous HLH cases caused by genetic defects accompanied by skin involvement. Hyperpigmentation, edema, erythema nodosum, maculopapular rashes, annular erythema, jaundice, and pustules are unusual skin manifestations of genetic defects related to HLH. Grisccelli syndrome type 2 and Chediak–Higashi syndrome are specific subtypes characterized by cutaneous hyperpigmentation, while cutaneous manifestations are nonspecific to other subtypes. In addition, the histologic appearances were nonspecific and rarely demonstrated cutaneous hemophagocytosis. The typical histological features are described as a lymphohistiocytic perivascular infiltrate in the reticular dermis, without evidence of epidermal changes or vasculitis, which is consistent with the skin biopsy in our case [45]. Hence, it is necessary to perform a marrow aspiration to exclude the possibility of HLH when various skin manifestations are incurable with routine treatments.

## 5. Conclusions

This study presents the first HLH case with stirred-up isolated skin lesions following COVID-19 vaccination with the background of a genetic defect of UNC13D. Without suggestive tests, the nonspecific and isolated lesions made it dramatically difficult to direct the diagnosis of HLH at an early stage. Clinicians should be aware of the possibility of HLH when unclassified skin lesions cannot be cured with routine treatments, and a marrow aspiration may be necessary to make a definite diagnosis. The adverse effects of COVID-19 vaccination should be made known, and full knowledge of the side effects caused by the vaccine should be analyzed and summarized. Individuals with preexisting systemic conditions should be cautious about vaccination and be closely followed up for any suspicious symptoms after vaccination. In the future, more cases should be included in multicentric trials to establish a formal clinical pathway, as in this thesis.

## Figures and Tables

**Figure 1 vaccines-10-01436-f001:**
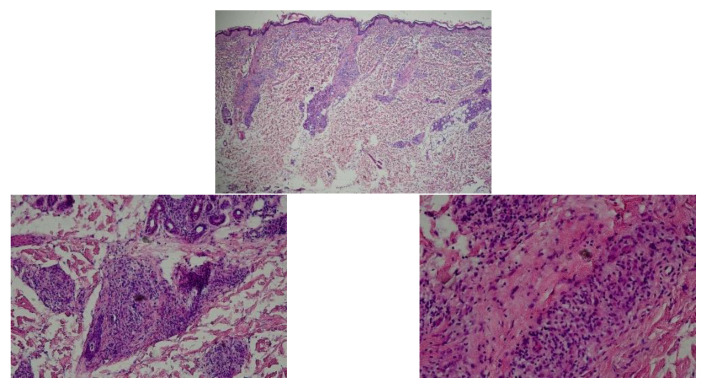
Skin biopsy revealed perivascular and periadnexal infiltration comprising histocytes and lymphocytes in the dermis, with the obvious involvement of nerves.

**Figure 2 vaccines-10-01436-f002:**
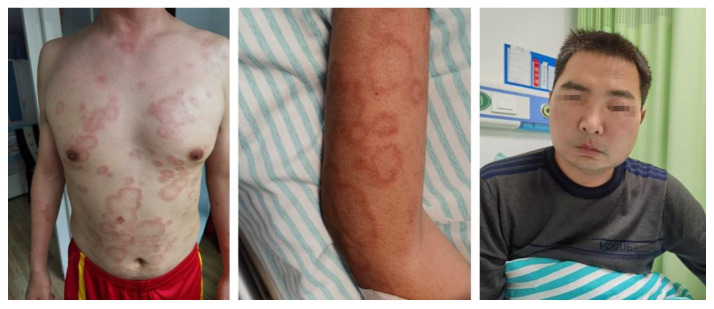
Generalized pruritic annular to irregular erythema, accompanied by edema on the left side of the face.

**Figure 3 vaccines-10-01436-f003:**
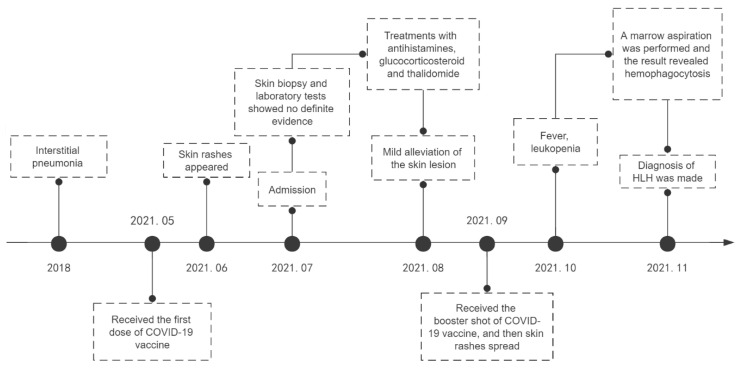
A coherent timeline indicating the date of this case.

**Figure 4 vaccines-10-01436-f004:**
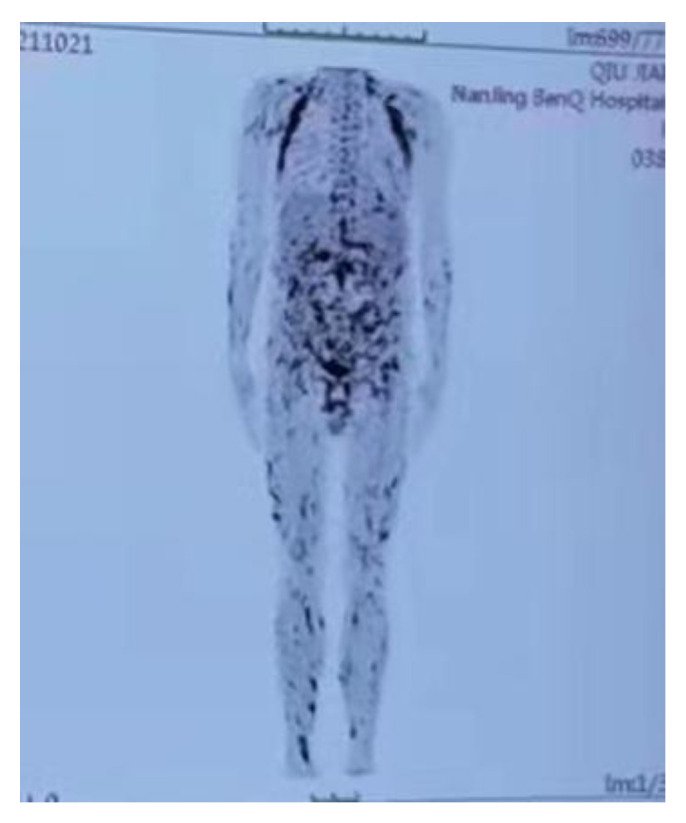
PET/CT revealed widespread increased metabolic activity.

**Table 1 vaccines-10-01436-t001:** Laboratory tests on the second admission.

Tests on Admission	Results	Normal Ranges	Tests on Admission	Results	Normal Ranges
Complete blood count			Hepatic and renal function		
White blood cell (10^9^/L)	**2.09**	3.5–9.5	ALT (U/L)	27.8	5–46
Neutrophil count (10^9^/L)	**1.50**	1.8–6.3	AST (U/L)	**66.7**	8–40
Lymphocyte count (10^9/^L)	**0.43**	1.1–3.2	Lactate dehydrogenase (U/L)	**668**	100–245
Hemoglobin (g/L)	147.0	130–175	Serum creatinine (umol/L)	69.3	35–115
Platelet (10^9^/L)	**111**	125–350	Blood urea nitrogen (mmol/L)	4.97	2.9–8.2
Coagulation			Fasting triglycerides (mg/dL)	1.96	0.46–2.25
APTT (s)	34.00	25–37	Ferritin (ng/mL)	**18,669.00**	24.00–336.2
PT (s)	11.80	9–13.5	Soluble IL-2 receptor (U/mL)	**103,915**	<6400
Thrombin time (s)	**18.80**	10.3–16.6	NK cell (%)	**11.54**	≥15.11
Fibrinogen (g/L)	**1.83**	2–4			
D-Dimer (mg/mL)	322.00	0–550			

Abnormal results are shown in BOLD. APTT: activated partial thromboplastin time; PT: prothrombin time.

**Table 2 vaccines-10-01436-t002:** Reported cases of HLH following COVID-19 vaccination.

Case	Sex, Age	Symptoms Onset after Vaccination	Clinical Manifestations	Medical History	Other Cause of HLH	Gene Mutations	Treatment	Outcome
Our case	M, 38 y	4 weeks	Annular erythema, fever, fatigue	Interstitial lung disease	None	UNC13D	Methylprednisolone, etoposide, doxorubicin, HSCT	Improved and under follow-up
Tang et al., J Hematol Oncol, 2021 [6]	M, 43 y	Shortly after vaccination	Fever, vomiting, malaise	None	EBV infection	Absent	Dexamethasone	Discharged
Hieber et al., Infection, 2022 [7]	F, 24 y	10 days	Fatigue, fever, chills, nausea	None	None	Not tested	IVIG, dexamethasone, Anakinra	Discharged
Ai et al., J Med Virol, 2022 [8]	M, 68 y	10 days	Fevers, rigors, lethargy, night sweats	Hypertension, gout, Bowen’s disease	None	Not tested	No therapy	Spontaneous improvement
Nasir et al., J Clin Transl Res, 2022 [9]	M, 46 y	3 weeks	Fever, fatigue, disturbed sleep, reduced appetite, skin rashes, oral ulcers	None	None	Not tested	Dexamethasone	Improved
Caocci et al., Haematologica, 2021 [10]	M, 38 y	21 days	Fever, chills, fatigue, erythematous papules	None	None	Not tested	Methylprednisolone	Fully recovered
Baek et al., Infect Dis (Lond), 2021 [11]	M, 20 y	2 days	Fever, myalgia, nausea, skin rashes	None	Not provided	Not tested	Dexamethasone	Immediate improvement
	F, 71 y	7 days	Fever, neurologic symptoms	Hypertension	Not provided	Not tested	Dexamethasone, etoposide	Discharged
Sassi et al., Br J Haematol, 2021 [12]	M, 85 y	Shortly after vaccination	Anorexia, asthenia, pruritus	None	Not provided	Not tested	Not provided	Not provided
Rocco et al., Clin Infect Dis, 2021 [13]	M, 52 y	1 day	Fever, abdominal pain	A viral syndrome	T-cell lymphoma, EBV viremia	Not tested	Dexamethasone, etoposide	Death
	M, 53 y	4 days	Fever, worsening hypoxia	Interstitial lung disease	EBV viremia	Not tested	Dexamethasone, Anakinra, IVIG, rituximab	Discharged to rehab facility
	M, 57 y	12 days	Malaise, nausea	Heart failure, HIV infection, Mycobacterium avium, KSHV viremia	Kaposi sarcoma herpesvirus viremia	Not tested	Methylprednisolone	Death
	F, 55 y	3 days	Fever	Adult-onset Still’s disease, pulmonary aspergillosis, GATA2 deficiency	Not found	Not tested	Anakinra	Slowly recovered
	F, 48 y	4 days	Fever, cough, pleuritic chest pain	HIV	Not found	Not tested	Prednisone, infliximab	Improvement within 72 h
Attwell et al., J Clin Pathol, 2022 [14]	M, 65 y	5 days	Breathlessness, fever, myalgia	Type 2 diabetes mellitus	Not found	Not tested	Methylprednisolone, IVIG, Anakinra	Deteriorated
	F, 75 y	7 days	Night sweats, breathlessness, myalgia, fever, cough	JAK2-mutation positive essential thrombocythaemia, breast cancer	Not found	Not tested	Methylprednisolone, IVIG, Anakinra	Death
	M, 35 y	8 days	Fever, diarrhea, sore throat, pruritic rash, breathlessness	Ankylosing spondylitis	Not found	Not tested	Methylprednisolone	Responded well
Wu et al., BMJ Case Rep,2022 [15]	M, 60 y	6 days	Fevers, night sweats, loss of appetite, delirium, non-ambulatory	Barrett’s esophagus	Not found	Not tested	Dexamethasone, etoposide	Discharged, but relapsed and deteriorated
	F, 32 y	4 weeks	Fever	None	Not found	Not provided	Dexamethasone, etoposide, emapalumab-lzsg	Discharged, but deteriorated
Lin et al., Vaccines, 2022 [16]	F, 14 y	15 days	Fever, headache, nausea, tachypnea, drowsy consciousness, mottling skin, jaundice	None	EBV viremia	Not tested	Methylprednisolone, IVIG	Discharged
Cory et al., Clin Med (Lond), 2021 [17]	F, 36 y	9 days	Fever, myalgia, sore throat, mild facial swelling	None	Not found	Not tested	Methylprednisolone, IVIG	Improvement

M, male; F, female; y, years; IVIG, intravenous immunoglobulin; HSCT, hematopoietic stem cell transplantation.

**Table 3 vaccines-10-01436-t003:** Primary HLH with cutaneous manifestations.

Case	Sex, Age	Gene Defects	HLH Type	Initial Symptoms	Cutaneous Manifestations	Skin Histologic Findings	Treatments	Outcome
Minocha et al., Intractable Rare Dis Res, 2017 [18]	M, 20 m	RAB27A	Griscelli syndrome type 2	Fever, jaundice, pallor, weight loss	Icterus, silvery-gray hair, and hypopigmented skin	Not provided	Mycophenolate mofetil, HSCT	Improved and under follow-up
Mishra et al., Korean J Pediatr, 2014 [19]	M, 5 y	RAB27A	Griscelli syndrome type 2	Fever, skin lesion	A generalized hypopigmented skin and multiple erythematous to hyperpigmented, nodular lesions, extending from midthighs to feet	Widening of septae in the subcutaneous tissue, infiltration of the periphery of the fat lobule by chronic inflammatory cells	Prednisolone, HSCT	Not provided
Tewari et al., Spec Care Dentist, 2018 [20]	M, 4 y	RAB27A	Griscelli syndrome type 2	Pain in the oral cavity and tooth decay	Silvery white hair and white skin	Not provided	Not provided	Not provided
Panigrahi et al., Pediatr Neurol, 2015 [21]	F, 1 y	RAB27A	Griscelli syndrome type 2	Fever	Silvery white hairs, silvery eyelashes	Not provided	Methylprednisolone	Died
Meschede et al., Braz J Med Biol Res, 2008 [22]	M, 3 y	RAB27A	Griscelli syndrome type 2	Fever	Light silvery-gray colored scalp hair and eyebrows	Not provided	Corticoid, cyclosporine	Died
Gotesman et al., Pediatr Dermatol, 2020 [23]	F, 18 m	RAB27A	Griscelli syndrome type 2	Skin lesion	Non-pruritic, erythematous-violaceous papules, and dry, coarse, silvery-gray hair	A granulomatous inflammatory process	HSCT	Improvement
Guo et al., Medicine, 2018 [24]	M, 13 m	XIAP	X-linked lymphoproliferative syndrome	Fever, skin lesion, recurrent ear discharge	Widespread hemorrhagic skin eruptions	Langerhans cell histiocytosis (LCH)	HLH-2004- directed chemotherapy	Died
Kaya et al., Pediatr Blood Cancer, 2011 [25]	F, 4 y	LYST	Chediak–Higashi syndrome	Fever, pallor, lethargy, poor appetite	Speckled hypopigmented areas	Not provided	HLH-2004 therapy	Complete remission
Nielsen et al., J Pediatr Hematol Oncol, 2015 [26]	F, 2 m	LYST	Chediak–Higashi syndrome	Coryza, coughing, skin eruption	A pustule skin eruption, fair hair, pale and wax-like skin	Numerous large inclusion bodies in mast cells, and an epidermis virtually absent of melanin in both melanocytes and keratinocytes	Etoposide, dexamethasone, cyclosporine, IVIG	Died
Wu et al., Braz J Med Biol Res, 2017 [27]	M, 9 m	LYST	Chediak–Higashi syndrome	Fever	Mild pallor, gray hair, patchy hypopigmentation of the skin, red rashes on the trunk	Not provided	Cyclosporine A, dexamethasone, etoposide; HSCT has been planned	Temporary remission
Morrone et al., Case Rep Med, 2010 [28]	F, 16 m	CHD1	Chediak–Higashi syndrome	Fever, decreased activity, increased sleepiness, irritability	Silvery hair, pale skin, edematous eyelids	Not provided	Etoposide, dexamethasone, cyclosporine	Died
Sheng et al., BMC Med Genet, 2019 [29]	F, 30 y	STXBP2, LYST	Not provided	Fever, fatigue	Oedematose swelling of the face and coexistent skin lesions	Not provided	Etoposide, dexamethasone, currently waiting for HSCT	Well-controlled for a month
Larson et al., Pediatr Dermatol, 2017 [30]	M, 6 d	PRF1	F-HLH type 2	Skin lesion	Multiple blue–purple violaceous nodules	An intact epidermis and an underlying dermal infiltrate of mononuclear cells	Not provided	Not provided
Viñas et al., Front Immunol, 2021 [31]	M, 16 m	STXBP2	F-HLH type 5	Skin lesion, fever, vomit, diarrhea, edema	An exacerbation of cutaneous Langerhans cell histiocytosis, edema	Not provided	Not provided	Not provided
Tang et al., Medicine, 2019 [32]	F, 9 y	STXBP2	F-HLH type 5	Fever	Ecchymosis and edema of the lower extremities	Not provided	HLH-2004-directed chemotherapy, HSCT	Died
Pasqualin et al., Ital J Pediatr, 2014 [33]	M, 11 y	STX11	F-HLH type 4	Fever, dyspnea	A warm, painful, indurated plaque with a brownish, hyperpigmented over the right thigh	Mixed septal and lobular inflammatory infiltrate of benign-appearing histiocytes, plasma cells and lymphocytes, and diffuse fat necrosis	Methylprednisolone, cyclosporine	Remission was sustained at 6-month follow-up
Chen et al., J Clin Pathol, 2007 [34]	F, 11 y	PRF1	F-HLH type 2	Fever, skin lesion	Indurated skin nodules over the left thigh	Lobular panniculitis with lymphocytic infiltration with occasional benign histiocytes showing hemophagocytosis	13-cis retinoic acid, prednisolone	Died
Akyol et al., J Pediatr Hematol Oncol, 2020 [35]	F, 21 m	UNC13D	F-HLH type 3	Fever, skin lesion	Widespread maculopapular rash	Not provided	Not provided	Not provided
Zengin et al., Am J Dermatopathol, 2021 [36]	M, 4 y	UNC13D	F-HLH type 3	Skin lesion	A widespread popular–pustular rash	A cup-shaped depression of the epidermis, which exhibited perforation with necrobiotic collagen. Necrobiosis with palisading histiocytes and lymphoplasmacytic inflammatory cell infiltration in the dermis	Dexamethasone, HSCT	Not provided
our case	M, 36 y	UNC13D	F-HLH type 3	Skin lesion	Widespread annular erythema, facial edema	Lymphohistiocytic infiltration in the dermis, as well as reactive hyperplasias of lymphoid tissue dominated by cytotoxic T-cells	Methylprednisolone, etoposide, doxorubicin, HSCT	Improved and under follow-up

y, years; m, months; d, days; HSCT, hematopoietic stem cell transplantation; IVIG, intravenous immunoglobulin.

**Table 4 vaccines-10-01436-t004:** Genetic defects associated with primary HLH.

Type of HLH		Defective Gene
Familial HLH (F-HLH)	F-HLH type 2	PRF1
	F-HLH type 3	UNC13D
	F-HLH type 4	STX11
	F-HLH type 5	STXBP2
Immuno-deficiency syndromes	Griscelli syndrome type 2	RAB27A
	Chediak–Higashi syndrome	LYST
	Hermansky–Pudlak syndrome type 2	AP3B1
EBV-driven	X-linked lymphoproliferative disorder type 1 (XLP-1)	SH2D1A
	X-linked lymphoproliferative disorder type 2 (XLP-2)	BIRC4
	IL2-inducible T-cell kinase deficiency	ITK
	CD27 deficiency	CD27
	X-linked immunodeficiency with magnesium defect (XMEN)	MAGT1

## Data Availability

Not applicable.

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
