# Peer review of "Adult-Onset Familial Hemophagocytic Lymphohistiocytosis Presenting with Annular Erythema following COVID-19 Vaccination"

_vaccines, 2022, doi:10.3390/vaccines10091436_

Round 1

Reviewer 1 Report

Dear Author!

I was asked to review the manuscript “Adult-onset familial hemophagocytic lymphohistiocytosis presenting with annular erythema following COVID-19 vaccination: a case report and literature review” submitted to Vaccines (ISSN 2076-393X), manuscript vaccines-1853133 by YiFan He et al.

The case report provides a description of a very unusual and extremely rare complication - HLH soon after Covid19 vaccination and aggravation of symptoms after booster shot. The case is very well assessed, including markers, cell typing and genetics and treated accordingly as well. The review of added similar cases is clear and well organized in tables with relevant data. My comments are:

1.       I did not find the Figures 1,2,3, 4 that have been addressed in the section of Case description

2.       I suppose, that the conclusion to perform genetic screening before vaccination against Covid19 in general is too strong and does not have a basis. In patients with undefined skin or other systemic conditions the vaccination may have severe sequela.  The presented patient did have skin lesions not classified and not diagnosed with HLH a month before vaccination. I would suggest to change the conclusion in accordance with the presented case and the literature data.  

Best regards

Reviewer

Author Response

Response to Reviewer 1 Comments

Dear reviewer!

We are really sorry for the delay of reply. Thank you for your careful review of my manuscript, entitled “Adult-onset familial hemophagocytic lymphohistiocytosis presenting with annular erythema following COVID-19 vaccination”. We are truly grateful to yours full comments and thoughtful suggestions. Based on these comments and suggestions, we have made careful revises. All changes made to the text are in red color. We hope the new manuscript will meet your standard. Below you will find our point-by-point responses to the comments.

  1. I did not find the Figures 1,2,3, 4 that have been addressed in the section of Case description.

Re: We have presented the Figures in the section of Case description. (line68-98, page 2; line153-159, page 4)

  1. I suppose, that the conclusion to perform genetic screening before vaccination against Covid19 in general is too strong and does not have a basis. In patients with undefined skin or other systemic conditions the vaccination may have severe sequela. The presented patient did have skin lesions not classified and not diagnosed with HLH a month before vaccination. I would suggest to change the conclusion in accordance with the presented case and the literature data.

Re: We have proposed a modified conclusion that individuals with preexisting systemic conditions should be cautious about vaccination and be closely followed up for any suspicious symptom after vaccination. Please see the part of Conclusions for details. (line247-251, page 6; line286-292, page 7)

Thank you and best regards,

Sincerely,

YiFan He.

Reviewer 2 Report

General comments
The authors present the first case report on annular erythema following SARS CoV-2 vaccination in a hemophagocytic lymphohistiocytosis patient having confirmed unc 13 D protein mutation (UNC12D). HLH 2004 diagnostic critieria was applied. They have documented previous HLH case reports following COVID-19 vaccines in Tables. The review of literature shows that this case is unique in having confirmed pre-disposing genetic defect.

This rare HLH case with confirmed UNC12D is interesting and important for pro-active  awareness HLH immunological dysfunction and its risk factor for adverse events following COVID-19 immunizations.

The manuscript requires a major overall of English grammar and syntax-: Sentence structure and the content are confusing and difficult to follow. (see corrections in abstract as example)

Unfortunately, the clinical description and the laboratory findings in the case report are not presented in a logical and systematic fashion.

Tables are found in methods, results and discussion sections. A schematic diagram of this case report chronology is recommended to simplify and make coherent to the reader. The literature review cases should be in the results section only and information consolidated.

Specific
lines 64-71. cutaneous lesions spreading and edema on left side of the face after the booster vaccine shot seems to be the main clinical evidence of vaccine adverse event. the chronology of the patient symptoms, their underlying interstitial lung disease and resolution of cutaneous and nerve pathology is fragmented and incomplete. A coherent time line in a Figure indicating dates for this case is needed.

IgE allergic eosinophilia in adult annular erythema is rare but not mentioned or discussed as a possiblility? EAE should be investigated and ruled out as the cause of cutaneous involvement?

lines 67-68. the leukocytopenia has typo i.e.2.09 X 109/L (should be WBC 2 x 109 /L).

line 68-69. The “bone marrow aspirate revealed hemophagocytosis” cryptic. A complete pathology report with precise data is required.

line 69-71 “meanwhile he had dramatically elevated” soluble serum CD25 and ferritin and decreased NK function. The values of sIL2R and ferritin and other important immune parameters are missing. The patient values must be cited with the normal reference range (mean 95% CI) in a Table along with the complete patient hematology and routine clinical chemistry report.

line 58. Elevated ANCA, ANA and dsDNA were not found in this case... what about other autoantibodies of systemic and organ specific autoimmunity?

line 59-60 “given the involvement of nerves in the pathologic finding the possibility of leprosy was taken into account” The authors then cite negative Myobacterium by pcr testing.

Are there neurological or muscular manifestations in this case? Is an autoimmune etiology suspected? If so, a complete autoantibody work up is warranted. The authors are referred to numerous studies on autoantibodies in COVID-19 and in cases following SARS CoV-2 vaccination.

2021 Dotan A, Müller S, Shoenfeld Y et al. The SARS-CoV-2 as an instrumental trigger of autoimmunity, Autoimmun Rev. Apr; 20(4): 102792
2022 Yue Chen,Zhiwei Xu,Peng Wang,Xiao-Mei Li,Zong-Wen Shuai,Dong-Qing Ye,Hai-Feng Pan.New-onset autoimmune phenomena post-COVID-19 vaccination. Immunology. https://doi.org/10.1111/imm.13443
2022 Dodig D, Fritzler MJ, Naraghi A, Tarnopolsky MA, Lu JQ. Immune-mediated necrotizing myopathy afterBNT162b2 vaccination in a patient with antibodies againstreceptor-binding domain of SARS-CoV-2 and signalrecognition particle. Muscle & Nerve.2022;65:11–18

Author Response

Response to Reviewer 2 Comments

Dear reviewer!

We are really sorry for the delay of reply. Thank you for your careful review of my manuscript, entitled “Adult-onset familial hemophagocytic lymphohistiocytosis presenting with annular erythema following COVID-19 vaccination”. We are truly grateful to yours full comments and thoughtful suggestions. Based on these comments and suggestions, we have made careful revises. All changes made to the text are in red color. We hope the new manuscript will meet your standard. Below you will find our point-by-point responses to the comments.

  1. The manuscript requires a major overall of English grammar and syntax-: Sentence structure and the content are confusing and difficult to follow. (see corrections in abstract as example)

Re: The paper has been carefully revised to improve the grammar and sentence structure by a native English speaker. The corresponding suggestions mentioned above were revised.

  1. Unfortunately, the clinical description and the laboratory findings in the case report are not presented in a logical and systematic fashion.

    Re: We have revised the clinical description and the laboratory findings in a logical and systematic fashion in the section of case report. (line49-153, page 2-4)

  1. Tables are found in methods, results and discussion sections. A schematic diagram of this case report chronology is recommended to simplify and make coherent to the reader. The literature review cases should be in the results section only and information consolidated.

Re: We combined the part of Methods with Results to make the literature review straightforward, and Tables was only showed in the section of Literature review. (line160, page 4)

  1. lines 64-71. cutaneous lesions spreading and edema on left side of the face after the booster vaccine shot seems to be the main clinical evidence of vaccine adverse event. the chronology of the patient symptoms, their underlying interstitial lung disease and resolution of cutaneous and nerve pathology is fragmented and incomplete. A coherent time line in a Figure indicating dates for this case is needed.

Re: Thank you for this suggestion. The patient suffered from interstitial pneumonia 3 years ago. Skin rashes appeared after the first dose of COVID vaccination and spread accompanying with an edema on left side of the face after the booster vaccine shot. One month later, new accompanying symptoms including a high fever and fatigue appeared. A coherent time line was showed in a Figure 2. (line112, page 3)

  1. IgE allergic eosinophilia in adult annular erythema is rare but not mentioned or discussed as a possibility? EAE should be investigated and ruled out as the cause of cutaneous involvement?

Re: EAE has been ruled out basing on the normal levels of eosinophilic granulocyte and IgE(line54-61, page 2).

  1. lines 67-68. the leukocytopenia has typo i.e.2.09 X 109/L (should be WBC 2 x 109 /L).

Re: We have revised the mistake. (line103, page 3)

  1. line 68-69. The “bone marrow aspirate revealed hemophagocytosis” cryptic. A complete pathology report with precise data is required.

Re: A detailed pathology report has been presented in lines 103-105, page 3.

  1. line 69-71 “meanwhile he had dramatically elevated” soluble serum CD25 and ferritin and decreased NK function. The values of sIL2R and ferritin and other important immune parameters are missing. The patient values must be cited with the normal reference range (mean 95% CI) in a Table along with the complete patient hematology and routine clinical chemistry report.

Re: Thank you for this suggestion. The complete patient hematology and routine clinical chemistry report has been summarized in Table 1. (line110, page 3)

  1. line 58. Elevated ANCA, ANA and dsDNA were not found in this case... what about other autoantibodies of systemic and organ specific autoimmunity?

Re: The patient has been tested with whole autoantibodies including ANCA, ANA and dsDNA,etc, The results were all within normal levels and were fully elaborated in line 58-61, page 2.

  1. line 59-60 “given the involvement of nerves in the pathologic finding the possibility of leprosy was taken into account” The authors then cite negative Myobacterium by pcr testing.
    Are there neurological or muscular manifestations in this case? Is an autoimmune etiology suspected? If so, a complete autoantibody work up is warranted. The authors are referred to numerous studies on autoantibodies in COVID-19 and in cases following SARS CoV-2 vaccination.

Re: The patient didn’t present any neurological or muscular manifestations. Meanwhile, the autoimmune etiology was ruled out basing on the normal levels of whole autoantibody profile.

Thank you and best regards,

Sincerely,

YiFan He.

Reviewer 3 Report

In the introduction section, I recommend extending the part about the genetic basis of the hemophagocytic lymphohistiocytosis (HLH) disease. For example, explain if it is autosomal recessive. 

It is not clear why this study is unique. Table 1 describes several cases of HLH following covid-19 vaccination. Highlight why this case report is novel with respect to the literature. 

In table 3, specify if autosomal recessive or dominant.

Specify the limitations of the study.

Author Response

Response to Reviewer 3 Comments

Dear reviewer!

We are really sorry for the delay of reply. Thank you for your careful review of my manuscript, entitled “Adult-onset familial hemophagocytic lymphohistiocytosis presenting with annular erythema following COVID-19 vaccination”. We are truly grateful to yours full comments and thoughtful suggestions. Based on these comments and suggestions, we have made careful revises. All changes made to the text are in red color. We hope the new manuscript will meet your standard. Below you will find our point-by-point responses to the comments.

  1. In the introduction section, I recommend extending the part about the genetic basis of the hemophagocytic lymphohistiocytosis (HLH) disease. For example, explain if it is autosomal recessive. 

Re: Thank you for this suggestion. We have extended the part about the genetic basis in introduction section (line 38-40, page 1).

  1. It is not clear why this study is unique. Table 1 describes several cases of HLH following covid-19 vaccination. Highlight why this case report is novel with respect to the literature. 

Re: The present patient was the first reported case with defined genetic defects among cases of HLH following covid-19 vaccination. Additionally, the nonspecific and isolated lesions, atypical and meaningless examinations made it dramatically difficult to direct diagnosis of HLH in earlier stage. We have analyzed and highlighted the novelty of the present case in discussions (line233-235, page 6) and conclusion part. (line283-286, page 7).

  1. In table 3, specify if autosomal recessive or dominant.

Re: Thank you for this suggestion. However, all subtypes of primary HLH follow autosomal recessive inheritance, so we didn’t add a column to specify if autosomal recessive or dominant in the table.

Reference:

Steen EA, Hermiston ML, Nichols KE, Meyer LK. Digenic Inheritance: Evidence and Gaps in Hemophagocytic Lymphohistiocytosis. Front Immunol. 2021 Nov 17;12:777851.

  1. Specify the limitations of the study.

Re: This study only presents a single case, more patients should be included in multicentric trials in future to establish a formal clinical pathway as for this thesis. This section was added it in conclusion part. (line292-293, page 7)

Thank you and best regards,

Sincerely,

YiFan He.

Round 2

Reviewer 2 Report

The date of first diagnosis should be added to the timeline 

Figure 3 schematic diagram.

The authors need to discuss the Ferritin (ng/mL) 18669.00

The >500 ng/mL ferrtin is a  HLH diagnostic criteria

and the median [95% CI] is reported 6556 [2402-11,734]

This case report has dramatically high ferritin elevation.

Besides the dsDNA and ANAs the Anti-phospholipid

autoantibodies should be investigated along with

catastrophic inflammation caused by macraphage activation.

Author Response

Dear reviewer!

We are truly grateful to your professional review work on our manuscript, entitled “Adult-onset familial hemophagocytic lymphohistiocytosis presenting with annular erythema following COVID-19 vaccination”. Below you will find our point-by-point responses to the comments.

  1. The date of first diagnosis should be added to the timeline Figure 3 schematic diagram.

Re: The figure 3 has been revised according to your advisable suggestion. (line120, page3)

  1. The authors need to discuss the Ferritin(ng/mL) 18669.00. The >500ng/mL ferritin is a HLH diagnostic criteria and the median [95% CI] is reported 6556 [2402-11734]. This case report has dramatically high ferritin elevation.

Re: We referred to some relevant literature and found the median serum ferritin is reported 25720 ng/mL [range, 626-684000] in adult-onset HLH patients. The number of patients having serum ferritin >10,000 ng/mL, >25,000 ng/mL, and >50,000 ng/mL was 54 (74%), 38 (52.1%), and 17 (23.3%) patients, respectively; A higher optimal cut-off of 16,000 ng/ml ferritin (sensitivity and specificity; 79% each, NPV; 98%) was reported to be a good discriminator between HLH and non-HLH hyperferritinemic states among a large series of adults.

References:

Otrock ZK, Eby CS. Clinical characteristics, prognostic factors, and outcomes of adult patients with hemophagocytic lymphohistiocytosis. Am J Hematol. 2015 Mar;90(3):220-4.

Sarangi R, Pathak M, Padhi S, Mahapatra S. Ferritin in hemophagocytic lymphohistiocytosis (HLH): current concepts and controversies. Clin Chim Acta. 2020 Nov;510:408-415.

  1. Besides the dsDNA and ANAs, the Anti-phospholipid autoantibodies should be investigated along with catastrophic inflammation caused by macrophage activation.

Re: In view of the normal level of D-Dimer (Table 1), and the absent presentation of thrombus in clinical features and histopathology findings (line 56, line 140), the diagnosis of anti-phospholipid autoantibody syndrome was not considered.

Thank you and best regards,

Sincerely,

YiFan He.